# Deglacial Subantarctic CO$_2$ outgassing driven by a weakened solubility pump

Yuhao Dai [1,4] ✉, Jimin Yu [2,1] ✉, Haojia Ren [3] & Xuan Ji[1]

The Subantarctic Southern Ocean has long been thought to be an important contributor to increases in atmospheric carbon dioxide partial pressure (pCO$_2$) during glacial-interglacial transitions. Extensive studies suggest that a weakened biological pump, a process associated with nutrient utilization efficiency, drove up surface-water pCO$_2$ in this region during deglaciations. By contrast, regional influences of the solubility pump, a process mainly linked to temperature variations, have been largely overlooked. Here, we evaluate relative roles of the biological and solubility pumps in determining surface-water pCO$_2$ variabilities in the Subantarctic Southern Ocean during the last deglaciation, based on paired reconstructions of surface-water pCO$_2$, temperature, and nutrient utilization efficiency. We show that compared to the biological pump, the solubility pump imposed a strong impact on deglacial Subantarctic surface-water pCO$_2$ variabilities. Our findings therefore reveal a previously underappreciated role of the solubility pump in modulating deglacial Subantarctic CO$_2$ release and possibly past atmospheric pCO$_2$ fluctuations.

The Southern Ocean is widely regarded as a crucial source of carbon dioxide (CO$_2$) to the atmosphere during glacial terminations because this region serves as a window for CO$_2$ exchanges between the atmosphere and the ocean interior[1–4]. In the Southern Ocean, prevailing southern hemisphere westerly winds drive upwelling of carbon-rich deep waters surrounding Antarctica, some of which are transported northward to the Subantarctic Zone (SAZ)[5,6]. The surface SAZ exposes the newly upwelled carbon-rich deep waters to the atmosphere enabling CO$_2$ outgassing, before these waters are entrained to form intermediate and mode waters[5–7].

Changes in the SAZ have been thought to be critical to deglacial atmospheric pCO$_2$ rises, with a contribution estimated to be around 40 ppm[8–11]. In the SAZ, CO$_2$ tends to escape to the atmosphere due to elevated surface-water CO$_2$ partial pressures (pCO$_2$) driven by high surface-water dissolved inorganic carbon (DIC) concentrations[12,13] associated with the newly upwelled deep waters surrounding Antarctica. CO$_2$ outgassing in the SAZ is somewhat alleviated by biologically driven carbon sequestration that exports carbon to depths in the form

of organic matter, a process called the biological pump[1,14,15]. In addition to this biological process, it is important to note that surface-water pCO$_2$ is further affected by CO$_2$ solubility determined by seawater temperature and salinity, a process known as the solubility pump[14,16]. Changes in the solubility pump have been shown by modeling studies to contribute substantially to atmospheric pCO$_2$ variability[2,17–19]. Everything else being equal, warming lowers CO$_2$ solubility in seawater and thus increases surface-water pCO$_2$ with an effect to cause CO$_2$ outgassing from the ocean to the atmosphere[16]. Surface-water pCO$_2$ increases with increasing salinity, but the salinity effect on the CO$_2$ solubility is generally smaller than the temperature effect in most regions[20] (Supplementary Fig. 2).

During glacial terminations, it has been proposed that the biological pump efficiency was lowered, driving up CO$_2$ outgassing in the SAZ[1,11,21–23]. The deglacial SAZ biological pump efficiency decline has been linked to reduced supplies of micronutrients such as iron via eolian lithogenic fluxes[10,11,24]. In this case, weakened biological pump would leave more carbon unused in the SAZ surface, raising

[1]Research School of Earth Sciences, The Australian National University, Canberra, ACT, Australia. [2]Pilot National Laboratory for Marine Science and Technology (Qingdao), Qingdao, China. [3]Department of Geosciences, National Taiwan University, Taipei, Taiwan. [4]Present address: Department of Geology, Lund University, Lund, Sweden. ✉e-mail: yuhao.dai@geol.lu.se; jiminyuanu@gmail.com

surface-water $pCO_2$ which would stimulate $CO_2$ outgassing[1]. On the other hand, as manifested by Southern Ocean temperature reconstructions[25–27], deglacial SAZ warming, in theory, should weaken the solubility pump, raise surface-water $pCO_2$, and promote $CO_2$ outgassing[16] (Supplementary Fig. 1). Existing reconstructions in the SAZ indeed show elevated surface-water $pCO_2$ during the last degla-ciation (~18-11 ka BP)[22,23]. Yet, it remains unknown regarding the respective roles of biological and solubility pump changes in affecting these past surface-water $pCO_2$ rises in the SAZ and by extension atmospheric $pCO_2$ changes.

Here, we systematically investigate the contributions of the bio-logical and solubility pumps to the SAZ surface-water $pCO_2$ changes in the modern ocean and during the last deglaciation. For the deglacial investigation, we have generated a surface-water $pCO_2$ record using a sediment core from the Southwest Pacific, paired with nutrient utili-zation efficiency and sea surface temperature (SST) reconstructions. We evaluate the relative roles of biological and solubility pumps in regulating deglacial surface-water $pCO_2$ changes at our site location. The same approach is further applied to published records at three additional SAZ locations. Moreover, we examine simulated early deglacial carbon cycle changes in an earth system model[28] to distin-guish effects of the two pumps during the early deglaciation. All our investigations suggest a strong solubility pump effect on the SAZ surface-water $pCO_2$ changes, urging a rethinking of mechanisms underlying deglacial $CO_2$ outgassing from the SAZ and, by extension, past atmospheric $pCO_2$ variabilities.

## Results
### Solubility pump in the modern SAZ
In the modern SAZ, both biological and solubility pumps strongly control spatial distribution and seasonal variability of surface-water $pCO_2$[12,13,29]. Regarding the spatial distribution, Fig. 1a shows small northward declines in annual mean surface-water $pCO_2$ within the SAZ, where nutrient (nitrate) is progressively utilized equatorward (Fig. 1b). Enhanced nutrient consumption would lower surface-water $pCO_2$, but this biological effect is compensated by the opposing influence of equatorward surface-water warming in the SAZ (Fig. 1c). Consequently, surface-water $pCO_2$ changes caused by nutrient consumption is largely balanced by decreasing $CO_2$ solubility, and the total northward surface-water $pCO_2$ decline is marginal. Regarding the seasonal varia-bility, Fig. 1d shows minimal monthly surface-water $pCO_2$ deviation from the annual mean levels in the SAZ. By contrast, seasonal changes in the biological and solubility pumps each cause surface-water $pCO_2$ to fluctuate ~40 ppm around the annual mean level (Fig. 1e, f)[12,29]. Because the strong (weak) biological pump occurs during warm (cold) seasons with low (high) $CO_2$ solubility, effects from the biological and solubility pumps generally cancel each other, leading to little seasonal surface-water $pCO_2$ variability. Combined, these spatial and temporal patterns suggest that the solubility pump plays a critical role, com-parable to that of the biological pump, in determining the SAZ surface-water $pCO_2$ fluctuations in the modern ocean. Next, we move on to explore the impact of the solubility pump on past surface-water $pCO_2$ changes, a topic that has rarely been studied by proxy reconstructions.

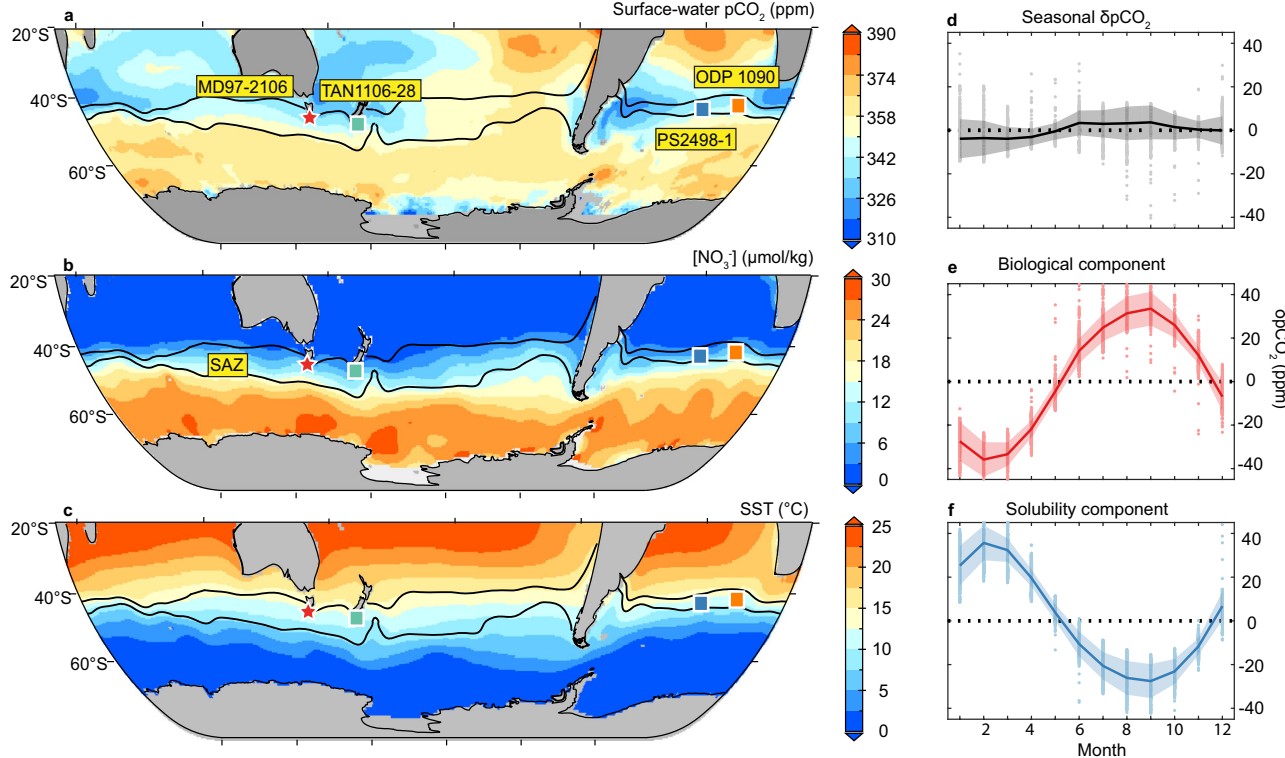

**Fig. 1 | Surface-water chemistry in the modern Southern Ocean. a** Annual mean surface-water $pCO_2$ during years 1985–2018. The map is made from data pre-sented in Gregor and Gruber[29]. https://creativecommons.org/licenses/by/4.0/. **b** Annual mean surface-water nitrate concentration. The map is made from data presented in Garcia et al.[48], accessible from https://www.ncei.noaa.gov/access/world-ocean-atlas-2018/. **c** Annual mean sea surface temperature. Two black curves indicate the modern positions of the Subtropical Front (STF; the northern curve) and the Subantarctic Front (SAF; the southern curve), respectively, and the region between them is the Subantarctic Zone (SAZ). The red star represents the location of our study site MD97-2106, and squares represent locations with published δ[11]B-based surface-water $pCO_2$ reconstructions in the SAZ. The map is made from data presented in Locarnini et al.[49], accessible from https://www.ncei.noaa.gov/access/world-ocean-atlas-2018/. **d** Monthly surface-water $pCO_2$ varia-bility within the SAZ (year 1985–2018) calculated from the OceanSODA-ETHZ dataset[29]. **e** Monthly surface-water $pCO_2$ variability attributed to biological pump changes within the SAZ calculated from the OceanSODA-ETHZ dataset[29]. **f** Monthly surface-water $pCO_2$ variability attributed to solubility pump changes within the SAZ calculated from the OceanSODA-ETHZ dataset[29]. In **d**–**f**, shadings show ±1σ standard deviation ranges of observations at discrete locations (represented by dots). As can be seen from **d**–**f**, the solubility pump plays a critical role in stabilizing SAZ surface-water $pCO_2$ by countering effects due to biological pump changes.

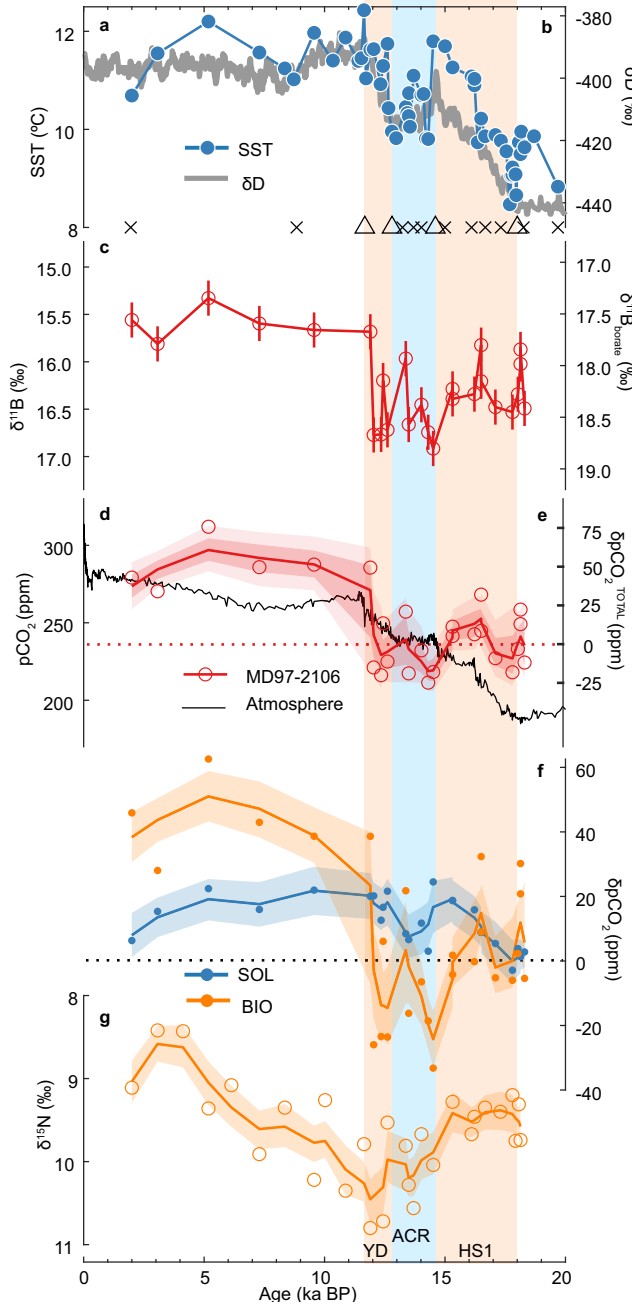

**Fig. 2 | Surface-water reconstructions at site MD97-2106. a** Sea surface temperature (SST) reconstructed from planktic foraminiferal Mg/Ca[36] (blue circles, left axis). **b** Antarctic temperature changes represented by δD[26] (gray curve, right axis). Crosses and triangles at the bottom are age control points based on planktic radiocarbon dating and SST-δD matching, respectively. **c** Planktic foraminiferal δ[11]B and seawater borate δ[11]B with error bars showing ±2σ uncertainties. **d** Reconstructed surface-water $pCO_2$ at site MD97-2106 (red circles) compared with atmospheric $pCO_2$ recorded in the Antarctic ice cores[43, 44] (left axis). **e** Total surface-water $pCO_2$ change relative to 18 ka ($\delta pCO_2^{TOTAL}$) at site MD97-2106 (right axis). **f** Deglacial surface-water $pCO_2$ attributed to the solubility and biological pumps ($\delta pCO_2^{SOL}$ and $\delta pCO_2^{BIO}$, blue and orange dots, respectively) at site MD97-2106. **g** Foraminifera-bound δ[15]N, a proxy reflecting surface nutrient utilization. Curves in **d**–**g** are derived from a LOESS smoother. In **d**, light and dark envelopes, respectively, represent 2.3–97.7% (roughly ±2σ) and 15.9–84.1% (roughly ±1σ) uncertainty ranges of timeseries incorporating uncertainties from measurements, all individual parameters used for calculations, and age models. In **f** and **g** only 15.9–84.1% uncertainty ranges are shown for clarify. The vertical pale orange bars represent Heinrich Stadial 1 (HS1; ~18.0–14.6 ka BP) and the Younger Dryas (YD; ~12.8–11.7 ka BP). The vertical pale blue bar represents the Antarctic Cold Reversal (ACR; ~14.6–12.8 ka BP). $\delta pCO_2^{SOL}$ and $\delta pCO_2^{BIO}$ (**f**) show similar structures to SST (**a**) and foraminifera-bound δ[15]N (**g**), respectively.

radiocarbon dates and tuning of SST at this site to Antarctic temperatures[26] (Fig. 2a, b; Supplementary Table 1, Supplementary Fig. 4). See Methods for analytical details.

Our reconstructed surface-water $pCO_2$ at site MD97-2106 fluctuated between ~210 and ~270 ppm during the last deglaciation, reached ~270 ppm at the onset of the Holocene, and increased by ~20 ppm from the early to middle Holocene (Fig. 2d). The range of deglacial surface-water $pCO_2$ at site MD97-2106 is comparable to previous reconstructions[22,23]. At our site, surface-water $pCO_2$ briefly dropped below the atmospheric $pCO_2$ values during the Antarctic Cold Reversal (ACR; 14.6–12.8 ka BP) and the Younger Dryas (YD; 12.8–11.7 ka BP) (Fig. 2d).

In addition to our surface-water $pCO_2$ reconstructions, we employ δ[15]N_{FB} to infer nitrate utilization efficiency at site MD97-2106. δ[15]N_{FB} reflects δ[15]N of surface-water nitrate taken up by foraminifera which increases as surface nitrate is progressively consumed by photosynthetic algae[11,37,38]. During Heinrich Stadial 1 (HS1; 18.0-14.6 ka BP) when the age model of our core is well constrained by a warming phase, δ[15]N_{FB} at our site remained roughly unchanged (Fig. 2b, g). This observation is consistent with a precisely dated coral-bound δ[15]N record from the same region showing minimal δ[15]N variability over the same period[21] (Supplementary Fig. 9). Maxima of δ[15]N_{FB} occurred during the ACR and the YD, coinciding with surface-water $pCO_2$ minima. During the Holocene, δ[15]N_{FB} declined by about 2.0‰, consistent with other fossil-bound δ[15]N records from the Southern Ocean[11,21,39–41].

## New deglacial SAZ surface-water pCO₂ and δ¹⁵N records

We present records of surface-water $pCO_2$ and nutrient utilization, respectively, based on boron isotopes (δ[11]B)[22,30,31] and foraminifera-bound nitrogen isotopes (δ[15]N_{FB}) of mixed-layer-dwelling planktic foraminifera species *Globigerina bulloides* for site MD97-2106 from the Southwest Pacific Ocean (Fig. 1). Site MD97-2106 (45.15°S, 146.28°E) is located in the northern part of the SAZ. Compared to today, the Subtropical Front, the northern boundary of the SAZ, possibly shifted northward to Tasmania during the Last Glacial Maximum (LGM; ~22-18 ka)[32,33]. During the last deglaciation, the Subtropical Front might migrate southwards[32,33], but unlikely to the south of our site. This is because the Subtropical Front marks a ~4 °C SST gradient[34], while reconstructions at our site only show ~3 °C SST change during the entire deglaciation[35,36]. Consequently, our site was likely located within the SAZ during the entire last deglaciation, ideal for investigating deglacial SAZ surface conditions. The age model of site MD97-2106 during the last deglaciation is based on new

## Evaluating past biological and solubility effects at site MD97-2106

We partition surface-water $pCO_2$ changes at site MD97-2106 into two components caused by changes in biological and solubility pumps, using a similar method previously applied to investigate modern surface-water $pCO_2$ variability[12,16,29]. Firstly, we calculate total in-situ surface-water $pCO_2$ changes (noted as $\delta pCO_2^{TOTAL}$) relative to the reference age of 18 ka; choosing a different reference age has little effect on our conclusions (Supplementary Fig. 6). Secondly, surface-water $pCO_2$ is recalculated using carbonate chemistry (i.e., DIC and alkalinity) fixed at the reference age, but using varying SST and sea surface salinity (SSS) based on our reconstructions (Methods). Thirdly, this recalculated surface-water $pCO_2$ is used to compute changes relative to 18 ka (the reference age). The relative surface-water $pCO_2$ changes calculated in this way are only driven by SST and SSS, and thus are attributed to the solubility pump effect (noted as $\delta pCO_2^{SOL}$). Fourthly, we calculate the difference between $\delta pCO_2^{TOTAL}$ and $\delta pCO_2^{SOL}$, which is defined as the

biology-driven surface-water $pCO_2$ change (noted as $\delta pCO_2^{BIO}$). We also provide an alternative approach to directly calculate $\delta pCO_2^{BIO}$, which yields consistent results with those presented in the main text (Methods; Supplementary Fig. 8). Any influence of external processes on surface-water $pCO_2$, such as changes associated with frontal shift, is embedded in our method, because these external processes affect surface-water $pCO_2$ via the carbonate chemistry, SST-SSS, or both. See "Methods" for calculation details.

As can be seen from Fig. 2f, $\delta pCO_2^{BIO}$ at site MD97-2106 fluctuated between ~−20 ppm and ~+40 ppm from the LGM to the early Holocene, followed by a ~20-ppm increase during the Holocene. From 18 to 15 ka, $\delta pCO_2^{BIO}$ showed little net change, suggesting marginal influence of biological processes on surface-water $pCO_2$ variations at our site. During the ACR and the YD, $\delta pCO_2^{BIO}$ exhibits mostly negative values, indicating a strengthened biological pump that would lower surface-water $pCO_2$ during these times. The evolution of $\delta pCO_2^{BIO}$ at our site is well corroborated by our $\delta^{15}N_{FB}$ record from the same core. Little net $\delta pCO_2^{BIO}$ change concurred with stable $\delta^{15}N_{FB}$ during HS1, while negative $\delta pCO_2^{BIO}$ during the ACR and the YD coincided with higher $\delta^{15}N_{FB}$ values which indicate more complete nutrient utilization (Fig. 2f–g). We note that, in addition to nitrate utilization in the SAZ, our $\delta^{15}N_{FB}$ might have also been affected by $\delta^{15}N$ of nitrate supplied to our site, which depends on the nitrate utilization of the nitrate source[11,21,41,42]. Despite these complications, similar deglacial structures in $\delta^{15}N_{FB}$ and independently derived $\delta pCO_2^{BIO}$ suggest that $\delta^{15}N_{FB}$ at site MD97-2106 reflects local nitrate utilization efficiency. Moreover, given their independent methods, consistent patterns of our $\delta pCO_2^{BIO}$ and $\delta^{15}N_{FB}$ lend strong support to our inference about past biological pump changes at this site. In stark contrast to previous findings in the SAZ[22–24], our $\delta pCO_2^{BIO}$ and $\delta^{15}N_{FB}$ suggest that the biological pump played a minor role in contributing to surface-water $pCO_2$ and thus air-sea $CO_2$ exchange at site MD97-2106 during HS1 and the YD when atmospheric $pCO_2$ rose substantially[43,44].

Compared with $\delta pCO_2^{BIO}$, our calculated $\delta pCO_2^{SOL}$, which reflects solubility pump effects, shows a different history (Fig. 2f). As expected, deglacial $\delta pCO_2^{SOL}$ increased in response to warming, although this warming effect was slightly counteracted by a declining SSS (Supplementary Fig. 5). From 18 to 10 ka, $\delta pCO_2^{SOL}$ showed a net increase of ~20 ppm, contributing to about one-third of the $\delta pCO_2^{TOTAL}$ at site MD97-2106 over the same period. More specifically, the $\delta pCO_2^{SOL}$ increase dominated the $\delta pCO_2^{TOTAL}$ rise from 18 to 15 ka, in contrast to little net change in the concurrent $\delta pCO_2^{BIO}$. This highlights the critical role of the weakened solubility pump in maintaining positive sea-air $pCO_2$ gradients, which would contribute to the atmospheric $pCO_2$ rise during HS1. During the YD, rising $\delta pCO_2^{SOL}$ helped reverse biological effects (shown by $\delta pCO_2^{BIO}$ and $\delta^{15}N_{FB}$) to limit the development of negative sea-air $pCO_2$ gradient (Fig. 2f-g), contributing to the contemporary atmospheric $pCO_2$ rise.

By separating influences of the biological and solubility pumps downcore, we demonstrate that a substantial portion of deglacial surface-water $pCO_2$ rise at our site originated from variations in the solubility pump. This suggests that the solubility pump, which has been neglected in previous investigations in the region, played an important role in regulating deglacial surface-water $pCO_2$ changes in the Pacific SAZ.

### Proxy and model data evaluation for broader SAZ

To quantify past influences of biological and solubility pumps in broader SAZ regions, we reanalyze deglacial surface-water $pCO_2$ changes at three additional locations using published proxy records[22,23] (Figs. 1a; 3b–d). At all the investigated sites, our calculated $\delta pCO_2^{BIO}$ suggests little-to-modest biological pump effects on SAZ surface-water $pCO_2$ during times with large atmospheric $pCO_2$ rises. For instance, except for a brief excursion at ~17 ka, $\delta pCO_2^{BIO}$ at Site ODP 1090 (Fig. 3b) varied little and suggests minimal biological pump

effects on surface-water $pCO_2$ during the last deglaciation[23]. At site PS2498-1, despite an overall larger $\delta pCO_2^{BIO}$ contribution to surface-water $pCO_2$ compared to other sites, roughly stable average $\delta pCO_2^{BIO}$ during ~14-11 ka contributed little to the observed surface-water $pCO_2$ rise (Fig. 3c). At site TAN1106-28, $\delta pCO_2^{BIO}$ showed a prominent peak at ~16 ka, but the general deglacial trend of $\delta pCO_2^{BIO}$ is poorly defined by the low temporal resolution. In contrast to $\delta pCO_2^{BIO}$ changes, our calculated $\delta pCO_2^{SOL}$ at all studied sites supports that solubility pump changes consistently contributed to deglacial $\delta pCO_2^{TOTAL}$. From 18 to 10 ka, we observe well-defined $\delta pCO_2^{SOL}$ increases of ~20 ppm at Site ODP 1090 from the South Atlantic. At site TAN1106-28 from the South Pacific, $\delta pCO_2^{SOL}$ increased by ~50 ppm, in response to the deglacial SST change of ~7 °C in part caused by a possible frontal shift over this site[23]. Although the record at site PS2498-1 does not cover the entire last deglaciation, $\delta pCO_2^{SOL}$ at this site shows a ~30-ppm increase during ~14−11 ka (Fig. 3c). These $\delta pCO_2^{SOL}$ changes significantly contribute to, or even dominate, the surface-water $pCO_2$ variations at these sites, strengthening our findings at site MD97-2106. Overall, our analyses of proxy data from different sectors of the Southern Ocean demonstrate that deglacial surface-water $pCO_2$ changes in the SAZ are substantially affected by solubility pump changes, rather than solely by biological pump changes as previously assumed[11,22,23].

We further scrutinize the role of the solubility pump in affecting deglacial SAZ surface-water $pCO_2$ in a simulation by climate model LOVECLIM[28]. In this simulation, a 30-ppm atmospheric $pCO_2$ increase is achieved during HS1 when the Southern Ocean overturning circulation and southern hemisphere westerly winds are intensified. The rising surface-water $pCO_2$ in the Southern Ocean is diagnosed as a main $CO_2$ source for the early deglacial atmospheric $pCO_2$ increase[28]. Using the same method applied to proxy data, we quantify $\delta pCO_2^{BIO}$ and $\delta pCO_2^{SOL}$ changes between 19 ka and 15 ka in this simulation ("Methods"). Our decomposition (Fig. 4) reveals that $\delta pCO_2^{BIO}$ changes are either small or tend to lower $pCO_2^{TOTAL}$ in the SAZ. Because nutrient utilization forced by iron availability is not prescribed in the model, our calculated $\delta pCO_2^{BIO}$ per se cannot be used to dismiss iron fertilization effect on deglacial SAZ surface-water $pCO_2$ changes. By contrast, $\delta pCO_2^{SOL}$ changed substantially due to strong surface warming, dominating surface-water $\delta pCO_2^{TOTAL}$ rise in the SAZ (Fig. 4). Therefore, our model data analyses suggest that solubility pump changes are crucial for deglacial surface-water $pCO_2$ and air-sea $CO_2$ exchange in the SAZ, strengthening our findings based on above extensive proxy reconstructions.

## Discussion

It is widely thought that declined biological pump efficiency, possibly owing to reducing nutrient utilization associated with dust-borne iron deposition, enhanced $CO_2$ outgassing in the SAZ and hence atmospheric $pCO_2$ rises during deglaciations[1,10,11,24]. During HS1 when atmospheric $pCO_2$ raised substantially and dust deposition in Antarctica declined dramatically[45] (Fig. 3), $\delta pCO_2^{BIO}$, together with independently measured $\delta^{15}N_{FB}$, indicates little net biological pump change at site MD97-2106. Over the same period, $\delta pCO_2^{BIO}$ shows little increase in response to the reduced iron fertilization inferred from dust deposition in Antarctica[45] and other examined sites. At two of the examined sites (ODP 1090 and PS2498-1), where opal and lithogenic fluxes are available[24,46–48], $\delta pCO_2^{BIO}$ shows correlation with neither flux during HS1[24,46,47] (Fig. 3b, c). In addition to local nutrient utilization efficiency, $\delta pCO_2^{BIO}$ and $\delta^{15}N_{FB}$ at a certain site in the SAZ may also be affected by nutrient supplies modulated by shifts of the Southern Ocean fronts[11,21,39,41,42]. Everything else being equal, poleward movements of the Subtropical and Subantarctic fronts and thus the SAZ during HS1 would reduce nutrient supply, with an effect to raise observed nutrient utilization efficiency, at any given location in the SAZ. During HS1, declining local nutrient utilization efficiency deduced from reduced iron fertilization and poleward front shifts would have

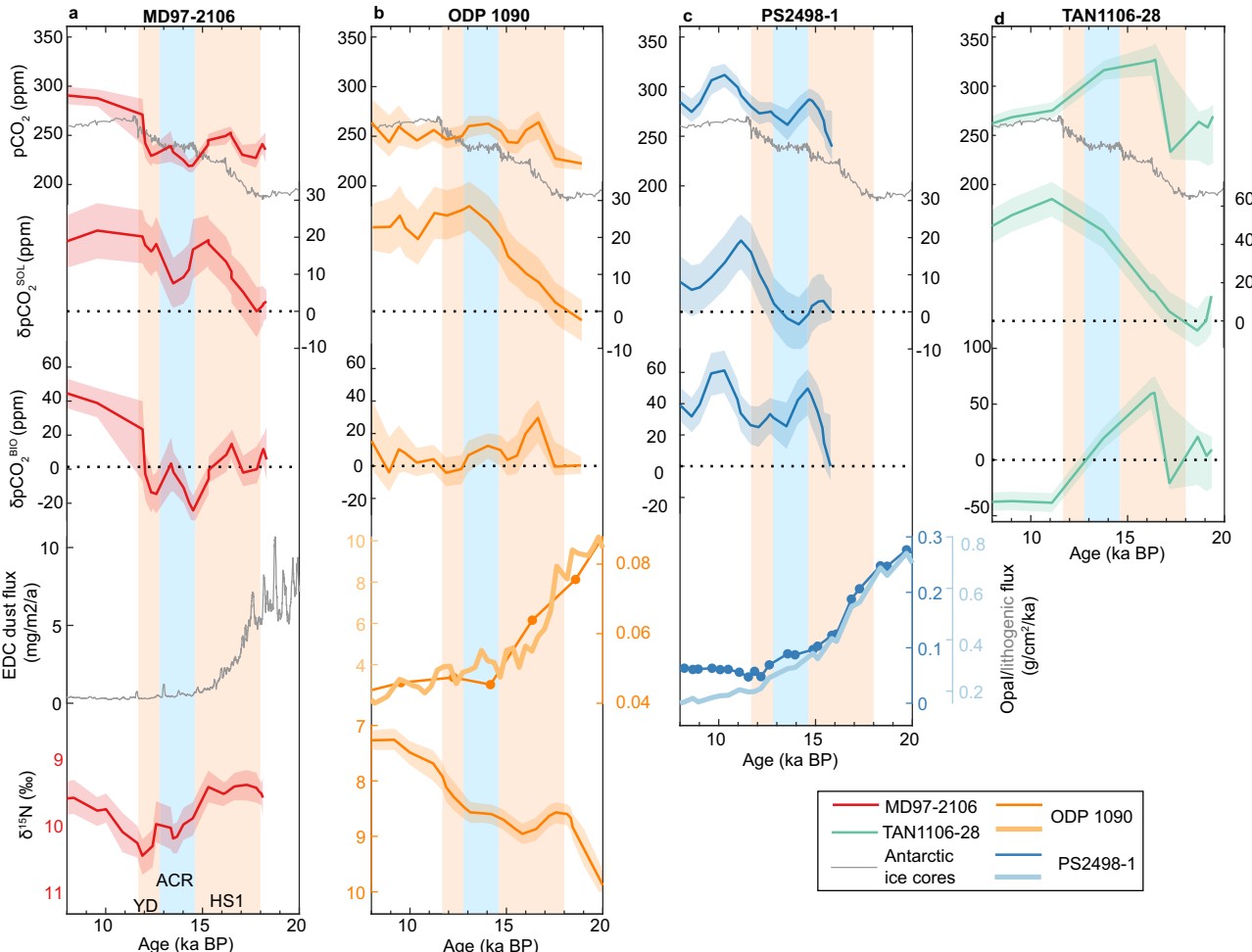

**Fig. 3 | Deglacial surface-water pCO₂, solubility and biological pump effects, dust and opal fluxes, and nutrient utilization at four Subantarctic Zone sites.** **a** MD97-2106; **b** ODP 1090[11, 23, 46]; **c** PS2498-1[22, 47]; and **d** TAN1106-28[23]. Data in the four panels are arranged as follows. First row: surface-water pCO₂ at investigated sites (red, orange, blue, and green curves) compared to atmospheric pCO₂ recorded in Antarctic ice cores[43,44] (gray curves). Second row: surface-water pCO₂ change attributed to the solubility pumps (δpCO₂ᴿˢᴼᴸ). Third row: surface-water pCO₂ change attributed to the biological pump (δpCO₂ᴮᴵᴼ). Fourth row: dust fluxes recorded in an Antarctic ice core[45] (**a**), lithogenic (pale orange curve)[22] and opal (orange curve with dots)[46] fluxes at site ODP 1090 (**b**), lithogenic (pale blue curve) and opal (blue curve with dots) fluxes at site PS2498-1[47] (**c**). Fifth row: Foraminifera-bound δ¹⁵N (this study and[11]). Note that scales of δpCO₂ᴿˢᴼᴸ and δpCO₂ᴮᴵᴼ differ in

**d** compared to **a**–**c**. Envelopes represent 15.9–84.1% uncertainties incorporating uncertainties from measurements, all individual parameters required for calculations, and age models. The vertical pale orange bars represent Heinrich Stadial 1 (HS1; -18.0–14.6 ka BP) and the Younger Dryas (YD; -12.8–11.7 ka BP). The vertical pale blue bar represents the Antarctic Cold Reversal (ACR; -14.6–12.8 ka BP). The reference age for relative surface-water pCO₂ change decomposition at these sites is set at 18 ka, except at site PS2498-1, where it is set at the oldest age of -15.8 ka. For all sites examined, solubility pump changes consistently contribute -20–50 ppm to the deglacial surface-water pCO₂ changes (second row). Surface-water pCO₂ changes attributed to the biological pump (third row) differ from dust and export production (fourth row).

opposing effects on SAZ δpCO₂ᴮᴵᴼ, and their combined effects may result in minimal δpCO₂ᴮᴵᴼ changes overall. Thus, despite the lack of any δpCO₂ᴮᴵᴼ decline, our reconstructions imply a potential role of iron fertilization in affecting surface-water pCO₂ through nutrient utilization in the SAZ during HS1.

Instead of a dominant biological contribution, our systematic investigations reveal persistent influences of the solubility pump on surface-water pCO₂ fluctuations in the SAZ under both modern and past conditions. Modern hydrographical data shows that the solubility pump causes surface-water pCO₂ to fluctuate by -40 ppm seasonally (Fig. 1). New and published proxy data demonstrates a solubility effect that can modulate surface-water pCO₂ by -20–50 ppm on millennial timescales during the last deglaciation (Figs. 2 and 3). The strong solubility pump influence is likely widespread in the SAZ, as shown by >20 ppm surface-water pCO₂ increase attributable to solubility pump changes during the early deglaciation in a model simulation[28] (Fig. 4). Notably, in this model simulation, solubility pump changes in the SAZ

lead to strong CO₂ outgassing and, together with other Southern Ocean processes, contribute to a full-scale atmospheric pCO₂ increase during HS1 without invoking nutrient utilization efficiency changes related to iron fertilization[28]. Based on our analyses on modern data, proxy reconstructions, and modeling outputs, we suggest that a weakened solubility pump, driven by warming, contributed critically to the rising surface-water pCO₂ in the SAZ and thereby maintaining this region as a CO₂ source to the atmosphere during the last deglaciation.

In sum, while our δpCO₂ᴮᴵᴼ reconstructions, at face value, indicate little biological pump contributions to deglacial SAZ surface-water pCO₂ variations at various SAZ sites locally, these reconstructions imply iron-related nutrient utilization effects on the deglacial SAZ surface-water pCO₂ variabilities when the influence of frontal shift is considered. Nevertheless, such an iron-related biological effect appears to be smaller than previously thought. Therefore, the common view that the iron-regulated SAZ biological pump changes substantially contributed to the deglacial atmospheric pCO₂ rise[8–11] may

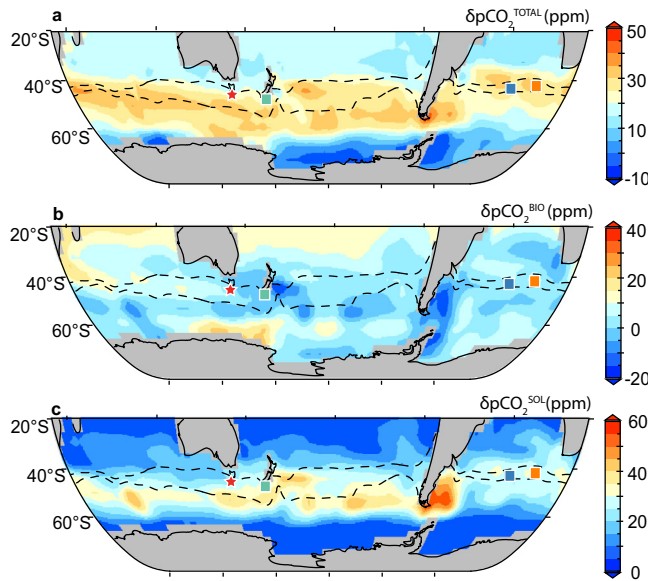

**Fig. 4 | Southern Ocean surface-water pCO₂ changes in a climate model simulation during the last deglaciation**[28]. **a** Total surface-water pCO₂ change ($\delta pCO_2^{TOTAL}$). **b** surface-water pCO₂ change attributed to the biological pump ($\delta pCO_2^{BIO}$). **c**, surface-water pCO₂ change attributed to the solubility pump ($\delta pCO_2^{SOL}$). All anomalies (δ) represent changes between 19 ka and 15 ka. Note different δpCO₂ scales for the three panels. The region between the dotted lines is the modern Subantarctic Zone (SAZ). The locations of MD97-2106 (red star) and other investigated sites (ODP 1090, orange square; PS2498-1, blue square; TAN1106-28 teal square) are shown. As can be seen, the total $\delta pCO_2^{TOTAL}$ rise in the SAZ from 19 to 15 ka in this model simulation is mainly driven by solubility pump changes. The figure is made from recalculation based on data presented in Menviel et al.[28].

need to be re-evaluated. In comparison to the widely recognized biological pump effect, the potential effect of the solubility pump in the SAZ has been previously overlooked when explaining the past atmospheric pCO₂ changes. Our work demonstrates that the solubility pump plays an indispensable role in modulating SAZ surface-water pCO₂ under both modern and past conditions. We suggest future works on quantifying the effect of SAZ solubility pump on past and possibly future atmospheric pCO₂ changes, which would have important implications for our mechanistic understanding of the global carbon cycle.

## Methods
### Trace element and boron isotope analyses
About 30 to 40 shells of planktic foraminifera *G. bulloides* from the 300−355 µm size fraction and >5 mg of *G. bulloides* shells from the 250−355 µm size fraction were picked for trace element and δ¹¹B analyses, respectively. These samples were cleaned following the "Mg-cleaning" procedure[50–53]. Measurements of B/Ca and Mg/Ca, along with Al/Ca and Mn/Ca for monitoring contaminants, were performed on an iCAP Inductively coupled plasma-mass spectrometry (ICP-MS) at the Australian National University (ANU), following an established method[51].

Separation of boron from sample matrices and measurement of δ¹¹B on a multi-collector-ICP-MS (MC-ICP-MS) generally follows the method of Foster[31] with some modifications. The cleaned foraminifera shells were dissolved in 0.5 M HNO₃, and buffered by 2 M NH₄Ac, instead of NaAc-HAc mixture, to pH of ~5.5. We changed the buffering solution to eliminate potential matrix contamination of Na on the δ¹¹B measurement. The buffered solution was gravitationally dripped into micro-columns, loaded with 20 µL ion exchange resin (Amberlite IRA-743, 63–125 µm size fraction), which was precleaned by 0.5 M HNO₃

and then boron-free deionized water. These micro-columns were tested by processing reference materials (boric acid solutions, NIST SRM 951 and ERM AE-121, without and with addition of CaCO₃ matrix; standard carbonates, NEP-3B) and generating values consistent with published values. After rinsing the resin eight times using Milli-Q water, boron was eluted by five aliquots of 90 µL 0.5 M HNO₃. A sixth aliquot was also added and collected to check for complete boron recovery. Total procedure blanks for each batch of samples were monitored, and were between ~20 and 100 pg.

δ¹¹B was measured on a MC-ICP-MS (Neptune Plus) at ANU using a standard bracketing method similar to Foster[29]. Following Farmer et al.[54], we measured boron blanks before every bracketing standard (NIST SRM 951) and sample. We also introduce water aerosol into the spray chamber through a second nebulizer after every standard/sample measurement in addition to the routine rinse, to flush out boron in order to minimize the memory effect of boron. An analytical block is as follows: flush-blank-standard-flush-blank-sample-flush-blank-standard. With the additional water flushing in between, measured blanks for ¹¹B can be kept <1.8% (an average for blanks of all the samples, external standards, and bracketing standards during 5 sessions) of the bracketing standard with 30 ppb of boron. Prior to and during analyses of these samples, repeating measurements of standard materials (NIST SRM 951, ERM AE-121, NEP-3B, and NIST RM 8301f) yield results consistent with their published values (Supplementary Table 2). The external reproducibility is estimated by repeating measurements of standard ERM AE-121 at 30-ppb boron concentration along with the samples (2σ = 0.17‰, *n* = 12). The boron concentration of the standard is chosen to match the expected median concentration of samples. Three of the foraminiferal samples were divided into two subsamples and processed separately from the cleaning step, and standard deviations of these replicated samples range from 0.08 to 0.27‰ (Supplementary Table 3).

During the late LGM and HS1, *G. bulloides* δ¹¹B we measured at site MD97-2106 agrees with previous measurements by MC-ICP-MS in the Pacific and Atlantic SAZ[23] and the Subtropical Southwest Pacific[55], but is on average ~1‰ lower than those from the same site measured on a Negative Thermal Ionization Mass Spectrometry (N-TIMS) by a previous study[56] (Supplementary Fig. 3). We tentatively attribute such offsets to potential analytical biases between MC-ICP-MS and N-TIMS that, as shown by a previous study, range from 0.5 to 2.7‰ and appear to enlarge for samples with low B/Ca values[54]. Despite that Moy et al.[56] show different deglacial δ¹¹B and thus surface-water pCO₂ magnitudes, deglacial $\delta pCO_2^{SOL}$ and $\delta pCO_2^{BIO}$ calculated using their data show similar patterns to those based on our new data (Supplementary Fig. 3).

### Carbonate chemistry system calculation
*G. bulloides* δ¹¹B is converted into δ¹¹B of seawater borate ($\delta^{11}B_{borate}$) using the calibration from Raitzsch, et al.[30]: $\delta^{11}B_{borate} = (\delta^{11}B_{G.\ bulloides} + 3.58 \pm 11.77)/(1.09 \pm 0.65)$. To estimate pH, SST and surface seawater salinity (SSS) are required. SST is estimated from *G. bulloides* Mg/Ca using the calibration of Elderfield and Ganssen[35]. SSS is estimated from the global sea-level change following Foster[31]. To calculate seawater pCO₂, seawater alkalinity is estimated from the modern seawater SSS-alkalinity relation[20]. We then use the CO2sys script[57] to calculate seawater pCO₂ and other carbonate chemistry parameters including DIC. The 2.3−97.7% uncertainties of seawater pCO₂ are propagated by a 10,000-iteration Monte-Carlo method incorporating uncertainties from δ¹¹B (2σ = 0.17‰), SST (2σ = 1 °C), and SSS (2σ = 0.5), and alkalinity which is sourced from SSS and the modern SSS-alkalinity relation[20]. Using a different way to estimate alkalinity (Supplementary Fig. 7) does not substantially affect our calculated seawater pCO₂ and its decompositions. For published δ¹¹B records, surface-water pCO₂ is recalculated using the same method as this study to be consistent with our methodology. Final uncertainties shown in Figs. 2 and 3 also

incorporate age uncertainties. For site MD97-2106, age uncertainty ($1\sigma$ ranging from 0.2 to 0.8 ka) is derived from the Undatable script[58], and for published records, a uniform age uncertainty ($1\sigma$) of 0.5 ka is assigned.

We partition the total in-situ surface-water $pCO_2$ changes ($\delta pCO_2^{TOTAL}$) into two components: solubility-driven ($\delta pCO_2^{SOL}$) and biology-driven ($\delta pCO_2^{BIO}$) components. In the main text, $\delta pCO_2^{SOL}$ is derived first, and the different between $\delta pCO_2^{TOTAL}$ and $\delta pCO_2^{SOL}$ is defined as $\delta pCO_2^{BIO}$. Here, we provide an alternative method to derive $\delta pCO_2^{BIO}$ first and subsequently $\delta pCO_2^{SOL}$. Firstly, $\delta pCO_2^{TOTAL}$ is calculate the same way as described in the main text. Secondly, we use DIC and alkalinity values (the same as those used for in-situ $pCO_2$ calculations), but constant SST and SSS values at 18 ka to calculate new surface-water $pCO_2$. It is important to note that DIC and alkalinity are used as intermediate parameters for calculations, and their values do not need to be accurately quantified to yield well-quantified new $pCO_2$ values. This is because DIC and alkalinity are inherently linked given constraint from pH (see Yu et al.[59] for detailed discussions). Thirdly, $\delta pCO_2^{BIO}$ is calculated by changes in the newly calculated surface-water $pCO_2$ relative to 18 ka. Fourthly, $\delta pCO_2^{SOL}$ is defined by differencing $\delta pCO_2^{TOTAL}$ and $\delta pCO_2^{BIO}$. As can be seen from Supplementary Fig. 8, the two methods yield consistent results, strengthening reliability of our calculation. The small differences between these two methods are due to the non-linear responses of seawater $pCO_2$ to temperature and DIC changes.

For $\delta pCO_2^{SOL}$, it may be further partitioned into temperature- and salinity-driven components ($\delta pCO_2^T$ and $\delta pCO_2^S$, respectively). We first calculate new surface-water $pCO_2$ at each age by using constant DIC, alkalinity, and SSS at 18 ka, but varying SST. Then, $\delta pCO_2^T$ is derived as changes of the newly calculated $pCO_2$ relative to 18 ka. Afterward, $\delta pCO_2^S$ is defined as the difference between $\delta pCO_2^{SOL}$ and $\delta pCO_2^T$. As can be seen from Supplementary Fig. 5, salinity changes tend to counter temperature effect on $pCO_2$, but $\delta pCO_2^S$ are limited to within 10 ppm at all four SAZ sites studied.

Regarding model outputs[28], we first calculate $\delta pCO_2^{TOTAL}$ between 15 ka and 19 ka, simply by differencing in-situ surface-water $pCO_2$ values at these times ($pCO_2^{in\text{-}situ,15ka}$ and $pCO_2^{in\text{-}situ,19ka}$, respectively). We re-calculate surface-water $pCO_2$ at 15 ka ($pCO_2^{recalc,15ka}$) using DIC and alkalinity at 15 ka but using SST and SSS at 19 ka. Similar to proxy data, $\delta pCO_2^{BIO}$ and $\delta pCO_2^{SOL}$ are calculated by: $\delta pCO_2^{BIO} = pCO_2^{recalc,15ka} - pCO_2^{in\text{-}situ,19ka}$ and $\delta pCO_2^{SOL} = pCO_2^{TOTAL} - \delta pCO_2^{BIO}$.

For modern hydrological data[29], monthly $\delta pCO_2^{TOTAL}$, $\delta pCO_2^{BIO}$, and $\delta pCO_2^{SOL}$ represent deviations from annual mean values. $\delta pCO_2^{TOTAL}$, $\delta pCO_2^{BIO}$, and $\delta pCO_2^{SOL}$ are calculated similarly to those for model results described above. For example, monthly $\delta pCO_2^{BIO}$ is calculated using monthly alkalinity and DIC but annual mean SST and SSS, while monthly $\delta pCO_2^{SOL}$ is calculated using annual mean alkalinity and DIC but with monthly SST and SSS (Fig. S2).

### Foraminifera-bound nitrogen isotope analyses

Sample preparation and measurements of $\delta^{15}N$ follow protocols in Ren et al.[60]. For each sample, >3 mg of *G. bulloides* shells (250–355 $\mu$m size fraction) were picked and crushed under a dissecting microscope. Foraminiferal samples were sonicated in an ultrasonic bath for 5 min with 2% polyphosphate solution, treated in bicarbonate-buffered dithionite–citric acid in an 80 °C water bath for 1 h, and added with basic potassium persulfate solution and autoclaved at 121 °C for 1 h. After every cleaning step, samples were rinsed with deionized water. Cleaned samples were dried overnight at 55 °C. Each sample was weighed (-1.5–3.5 mg) into a combusted glass vial and then dissolved in 3 M HCl to release organic N from the calcite shell. Persulfate oxidation reagent (POR, 0.3 g of 3-time-recrystallized basic potassium persulfate and 0.7 g of NaOH dissolved in 100 mL of deionized water) were added to the dissolved samples which were then autoclaved at 121 °C for 1 h to convert organic N to nitrate. The nitrate concentrations of all POR-

oxidized samples were measured to determine N contents after autoclaving using the chemiluminescence method[61]. Average N content of the cleaned calcite samples is 3.07 mmol N per gram. Nitrate concentration of POR and its $\delta^{15}N$ were constrained by two organic standards (US Geological Survey (USGS) 40, $\delta^{15}N = -4.5$‰ vs. air; and a laboratory standard, mixture of 6-aminocaproic acid and glycine, $\delta^{15}N = 5.4$‰ vs. air) processed along with samples. Nitrate concentration of POR is 0.2 $\mu$M, representing 1–3% of the total N in samples.

The denitrifier method was applied to transform dissolved nitrate and nitrite into nitrous oxide ($N_2O$) gas using a naturally occurring denitrifying bacterial strain, *Pseudomonas chlororaphis*, which lacks an active form of the enzyme $N_2O$ reductase. After degassing of the bacteria for 3 h, 1.5 mL of the bacterial concentrate was added with 5 nmol of samples acidified to pH of 3-7. Two nitrate reference materials (International Atomic Energy Agency $NO_3$ reference (IAEA-N3), $\delta^{15}N = 4.7$‰ vs. air; and USGS 34, $\delta^{15}N = -1.8$‰ vs. air) were processed along with samples to monitor the bacterial conversion and were later repeatedly measured between samples to check the stability of the mass spectrometry.

$\delta^{15}N$ of foraminiferal samples, together with bacterial blanks and organic standards, were determined by gas chromatography and isotope ratio mass spectrometry using a modified SigBench and MAT253 plus[62]. Due to the small sample size and low N content within foraminifera, no duplicates were made for these samples. Our IAEA-N3 and USGS 34 standards yielded standard deviation ($1\sigma$) of 0.06 and 0.07‰, respectively. The standard deviation ($1\sigma$) of the organic standards analyzed with these samples is 0.15‰, agreeing with the long-term variability of in-house carbonate standards using homogenized coral samples ($\pm 0.25$‰). As a result, we assume that the analytical error for the $\delta^{15}N_{FB}$ is 0.25‰ in our new record.

## Data availability

All data generated in this study have been deposited in the Zenodo database under access code: https://doi.org/10.5281/zenodo.6970032. All data generated in this study are also provided in the Supplementary Information.

## Code availability

Codes used to produce Figs. 1 and 4 are available from Y.D. upon request. Mode output data used to produce Fig. 4 is available from https://doi.org/10.4225/41/5af39aae7960f[28].

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

## Acknowledgements

We thank Brad Opdyke and Will Howard for help with arranging core materials, and Laurie Menviel for providing model outputs. This work is supported by NSFC42076056 and ARC Discovery Project DP190100894 to J.Y., and Ministry of Science and Technology, Taiwan (111-2636-M-002-020) to H.R.

## Author contributions

J.Y. designed the project. Y.D. derived the $pCO_2$ decomposition method. Y.D. made boron isotope measurements. X.J. made new trace element measurements. J.Y. supervised geochemical measurements made by Y.D. and X.J. H.R. made nitrogen isotope measurements. Y.D. performed data analyses and visualization. Y.D. wrote the first draft of the manuscript with significant inputs from J.Y. All authors contributed to the interpretation of the data and refinement of the manuscript.

## Funding

## Competing interests

The authors declare no competing interests.
