## [Peer Review File · Nature Communications]

REVIEWERS' COMMENTS

Reviewer #1 (Remarks to the Author):

The authors have paid close attention to the original reviewer comments and have answered them in detail with considered answers that will have taken both time and thought. They have taken our comments on board in edits to the manuscript. Overall, the changes made to the manuscript have clarified and improved the text greatly. Both the methods and impact of the findings are more directly communicated, without the latter being (previously probably unintentionally) over-amplified. This is a robust study, and the conclusions are of significance.

Please find our reviewer responses to the revisions made in the attached document. This is to allow the use of colour in the text to distinguish between original comments, revisions, and the new reviewer responses.

Dear Reviewers,

First of all, we greatly appreciate your constructive comments on our work. We have carefully considered your feedback and made changes where appropriate. We wish our revisions meet your satisfaction.

Please see below for our point-by-point responses to your comments.

Reviewers #1 & #2 (Joint remarks to the Author):

Dai et al., present compelling new evidence from paired boron and nitrogen isotope data from a sediment core located in the Subantarctic to show that the solubility pump, rather than the biological pump, most likely drove the local changes in the surface water pCO₂. This new evidence overturns some conventional thinking that the biological pump in the Subantarctic played a significant role in the deglacial carbon cycle.

The study is very well-designed, particularly the use of the triple constraints from Mg/Ca, boron isotopes and nitrogen isotopes. They also do an excellent job folding in the existing literature and datasets and the paper is generally well-structured and well-written.

We thank the reviewers for the positive comments.

Below we outline a few major and minor comments from outside the paleoceanographic community. In particular we have some questions/comments about the bigger picture implications of this study and some concerns about the transparency of the method to the reader.

We address the reviewers' two main concerns in detail below. Regarding the big-picture implications of our study, we have revised Introduction and Discussion sections to emphasize our contributions to the understanding of surface-water pCO₂ changes in the SAZ, a region commonly thought to be critical for deglacial atmospheric pCO₂ rises. Regarding our methodology, we have 1) replaced the old method with a more straightforward one and 2) provided step-by-step details for our surface-water pCO₂ deconvolution method both in the main text and Methods.

The authors have paid close attention to the reviewer comments and have answered them in detail with considered answers that will have taken both time and thought. They have taken our

comments on board in edits to the manuscript. Overall, the changes made to the manuscript have clarified and improved the text greatly. Both the methods and impact of the findings are more directly communicated, without the latter being (previously probably unintentionally) over-amplified. This is a robust study, and the conclusions are of significance.

Below are responses to each of the authors answers and edits, with a few minor suggestions for further improvement.

The authors have demonstrated that biological pump is not important locally in the Subantarctic, but does that mean the local solubility pump is important globally? Even if so, does it change our picture of the solubility pump as driver of atmospheric CO₂?

The authors conclude that “Therefore, we suggest that solubility pump changes in the subantarctic Southern Ocean should be considered as an important contributor when explaining past atmospheric pCO₂ fluctuations and projecting future atmospheric pCO₂ variabilities.”

The authors rightly stress the importance of the Subantarctic biological pump in setting global CO₂ levels. This zone of relatively high export feeds much of the deep ocean water masses through the sinking and eventual remineralisation of particulate organic matter. However, the solubility pump only feeds some of the intermediate waters, which themselves rather quickly outcrop in the tropics, where the solubility pump effect is then dominated by the dynamics of the tropics. Whilst not inconsequential, this is a much smaller water mass than the deep water masses formed in the polar regions. It is not clear to me if this would have a major impact on CO₂.

A quick back of the envelope calculation is as follows:

The authors show a 3 degree change in SST at their site. A 3 degree change in mean ocean temperature equates to about 30 ppm change. However, the volume of the ocean ventilated by the Subantarctic is relatively limited. Let's say a very generous maximum the upper 1000 meters or at most a ¼ of the global ocean. This would then equate to a change in CO₂ of at most 7.5 ppm. This isn't small number but nor is it a game-changing number. One could also use their estimates of pCO₂ change with solubility (~20 ppm). In this case, the global impact is even smaller.

We appreciate the reviewers' insightful comments on the big-picture impact of our findings on deglacial atmospheric pCO₂ changes. Our study is related to the following two critical but not fully resolved questions associated with the SAZ:

1. What mechanism(s) drove deglacial surface-water pCO₂ changes in the SAZ?
2. What magnitude can the SAZ changes contribute to deglacial atmospheric pCO₂ rises?

Regarding Question #1, a common view is that biological pump changes are responsible for deglacial surface-water pCO₂ variations in the SAZ (Hain et al., 2010; Jaccard et al., 2013; Martinez-Garcia et al., 2011). Regarding Question #2, based on the common view about the SAZ biological pump (the answer to question #1), previous studies suggest that the SAZ could cause up to ~40 ppm rise in atmospheric pCO₂ during deglaciations (Brovkin et al., 2007; Hain et al., 2010; Jaccard et al., 2013; Martínez-García et al., 2014). Evaluating the validity of answers to either question is critical to improve our understanding of the past global carbon cycle.

In this manuscript, we mainly focus on Question #1. Surface-water pCO₂ critically governs air-sea CO₂ exchange and thereby atmospheric pCO₂. Knowing reasons underlying surface-water pCO₂ variations in the SAZ can thus provide insights into mechanisms affecting atmospheric pCO₂. Based on extensive studies of modern, proxy, and model data, we demonstrate that 1) the biological pump effect is not as important as previously thought during the last deglaciation, and 2) the solubility pump is important for surface-water pCO₂ changes in the SAZ. We thus suggest rethinking of the potential role of the SAZ biological pump in modulating deglacial atmospheric pCO₂. Instead, the effect of solubility pump warrants consideration when evaluating SAZ surface-water pCO₂ changes. The significant solubility contribution to the SAZ deglacial surface-water pCO₂ changes challenges the long-held view that SAZ biological pump changes dominated the deglacial CO₂ outgassing from this region. Therefore, we feel our findings represent an important step to advance mechanistic understanding of oceanic carbon cycle and past atmospheric pCO₂ changes.

Regarding Question #2, we agree with reviewers that the small volume of the Subantarctic Mode Water (SAMW) can limit the contribution of the SAZ to the equilibrium oceanic carbon inventory. Nevertheless, it is worth noting that in addition to such an influence, air-sea CO₂ exchange in the SAZ kinetically affects the global carbon cycle. The SAZ exposed carbon-rich newly-upwelled deep waters from the Antarctic Zone, releasing carbon stored in the deep ocean to the atmosphere. In this process, the SAZ's influence on the atmosphere pCO₂ is not restricted by the SAMW volume, as CO₂ is released to the atmosphere before the SAMW formation. It is previously thought that during deglaciations, kinetic carbon loss through air-sea exchange is dominated by the biological pump change (i.e., weakened biological carbon sequestration), but we show that the positive seawater-atmospheric pCO₂ gradient is sustained by the solubility-induced surface-water pCO₂ rise instead during part of the last deglaciation. In another words, the carbon release to the atmosphere in the SAZ that is originated from the deep ocean is sustained by the declined CO₂ solubility during the last deglaciation. However, we acknowledge that our reasoning for such an effect is qualitative, and

more paleo and modelling studies are needed to better quantify the solubility pump contribution to atmospheric pCO₂ on the global scale.

It is possible that even taken both equilibrium inventory and kinetic effects into account, the SAZ's contribution to the deglacial pCO₂ rise may likely be smaller than previously thought. However, we here decide not to dwell too much on this point, but to focus on elucidating mechanisms of the SAZ surface-water pCO₂ change (Question #1). This is because our results may potentially change a longstanding view concerning Question #2, which requires more focused, meticulous, and extensive investigations as done in the main text.

In the main text, we have better stated our research aim (question #1) and shift the focus to discuss the solubility's impact in the SAZ in Abstract, Introduction, and Discussion (Lines 17-19,52-56, 229-231).

This consideration of the main questions of the study is valuable, and as the revised manuscript reads these are now much more apparent and consistent throughout.

'We hope the reviewers agree with us that addressing Question #2 warrants separate studies.'

This is reasonable, given the improved communication of the new text providing a clearer picture of the questions you aim to answer in the study. The omission of the need to investigate the global contribution further does not take away from the study giving conclusions of significance. Rather the findings that are presented now appear improved in their significant via clearer emphasis.

On a side note, possibly related to the comment about the referencing below, I am somewhat confused about how a 3 degree change in SST equates to only a 20 ppm rise in pCO₂.

Indeed, there is potential for some confusion here. We'd like to clarify that the effect is on pCO₂ changes of surface waters, not the atmosphere. Our calculations show that surface-water pCO₂ at site MD97-2106 increases by ~20 ppm in response to deglacial solubility pump changes including ~3 °C SST increase and a SSS decrease. In Fig. X1, we show that in the deglacial δpCO₂^{SOL} variability at our site, the SST contributed ~30 ppm, and the SSS contributed ~-10 ppm. The SST component changes are consistent with expectation based on the experiment-derived temperature sensitivity of 4.23% surface-water pCO₂ change per °C (Takahashi et al., 1993).

Fig X1, decomposition of $\delta p\text{CO}_2^{\text{SOL}}$ into $\delta p\text{CO}_2^{\text{T}}$ and $\delta p\text{CO}_2^{\text{S}}$ at site MD97-2106, using a reference age of 18 ka.

Ok, this is clarified.

Furthermore, we already know the global solubility pump fairly well from deep ocean temperature records and more recently mean ocean temperature records from ice cores (Bereiter et al., 2018). Most modelling studies already incorporate this information implicitly when they model the deglacial carbon cycle and the solubility pump remains fairly well-known (Köhler et al., 2005). That said, the authors may find some useful arguments if they look into the range predicted from models. For instance, the classic value is 30 ppm (e.g. Sigman and Boyle, 2000) but higher predictions have recently emerged at up to 40 ppm (e.g. Khatiwala et al., 2019). Overall, I would suggest the author's either refocus on their conclusions on the local impact or provide more evidence that their results significantly change our estimates of the global solubility. One may be able to support this by backing out the global impact of the solubility pump in their model results.

We follow the reviewers' valuable suggestion about refocusing on the SAZ. As stated above, our focus is to address mechanism(s) driving deglacial surface-water $p\text{CO}_2$ changes in the SAZ (Question #1) and we have clarified this point in the revision (Lines 17-19,52-56). Our new finding regarding the solubility pump is that solubility pump changes affect SAZ surface-water $p\text{CO}_2$ substantially during the last deglaciation. Based on our $\delta p\text{CO}_2^{\text{SOL}}$ reconstructions, we also suggest that the SAZ solubility pump can affect atmospheric $p\text{CO}_2$ via both ocean carbon storage and air-sea exchange in the SAZ before SAMW formation as previously discussed.

We also agree with the reviewers that the solubility influence on the global oceanic carbon inventory has been extensively studied by modelling studies. We now include more background from

modelling work in the Introduction (Lines 39-40) and extended discussions on related modelling works (Lines 195-209).

Again, the refocussing of the text on the SAZ specifically has presented a clearer story and negated the need to calculate global impacts, while not removing the significance of the findings. It is now reasonable to state that more detail on the global impacts to the carbon budget are beyond the scope of the study.

However, with the original comment in mind, it still doesn't seem quite right to say the solubility pump has been over-looked (L13, abstract). It has been included in these previous modelling studies, highlighted by all reviewers. Rather, it seems that it is the significance or potential contribution that has perhaps been underestimated. Or, do you say this for studies focussing on the SAZ very specifically? Could the authors re-phrase slightly to reflect whichever case?

Finally, on a small technical note, $p\text{CO}_2$ is not an accurate nomenclature for describing atmospheric CO_2 , as technically $p\text{CO}_2$ is a partial pressure and would require you to know atmospheric pressure. So it's very useful in oceanographic studies to discuss the local $p\text{CO}_2$ gradients between the surface ocean and the overlying atmosphere, but not a global indicator. Here is useful reference for how best to report atmospheric CO_2 (https://qml.noaa.gov/ccgg/about/co2_measurements.html)

We thank the reviewers for this useful comment. We agree with the reviewers that, in theory, atmospheric pressure and vapor pressure would affect air-sea CO_2 exchange, and that $p\text{CO}_2$ may not be the best notation when discussing modern observational data. However, the associated effects on seawater-atmospheric $p\text{CO}_2$ gradients are well within reconstruction errors associated with paleo studies. This is exactly the reason why the paleo-community has been routinely using $p\text{CO}_2$ for surface and atmospheric reconstructions.

The reviewers' concern may also stem from the use of " $p\text{CO}_2$ " for what. We have double-checked our manuscript to ensure that whenever " $p\text{CO}_2$ " is mentioned, it is always specified as surface-water $p\text{CO}_2$ or atmospheric $p\text{CO}_2$. In this way, we believe our nomenclature is accurate, when evaluating surface-water $p\text{CO}_2$, atmospheric $p\text{CO}_2$, or the air-sea $p\text{CO}_2$ gradients in the past.

In this manuscript, we used boron isotopes to reconstruct surface-water $p\text{CO}_2$, which is "the partial pressure of CO_2 in the gas phase that is in equilibrium with that seawater" (Zeebe & Wolf-Gladrow,

2001). This parameter is a seawater carbonate chemistry parameter determined by conservative parameters (dissolved inorganic carbon and alkalinity) and seawater physical properties (temperature, salinity, and pressure), and it is not related to atmospheric pressure. We also used atmospheric pCO₂ based on ice-core measurements, in combination with the surface-water pCO₂, to evaluate the air-sea pCO₂ gradient. In the case of ice-core atmospheric pCO₂, these measurements were calibrated to “dry standard air with known mole fractions calibrated at the NOAA Earth System Research Laboratory” (Marcott et al., 2014).

Additionally, we now use $\delta p\text{CO}_2$ to denote the relative change in surface-water pCO₂ at a certain site. We realize that $\Delta p\text{CO}_2$ is often used to denote the seawater-atmospheric pCO₂ gradient, and the old denotation in the previous version may cause confusion. We also changed notations for biological and solubility components ($\delta p\text{CO}_2^{\text{BIO}}$ and $\delta p\text{CO}_2^{\text{SOL}}$) for clarity.

The nomenclature is clarified and consistent throughout the manuscript. Just on a technicality – should the p of pCO₂ be italicised.

The data-model comparison is excellent but did the authors test the null hypothesis?

The null hypothesis in this case, would probably be that there were significant changes in nutrient utilization the Subantarctic forced by changes in the availability of iron. However, the Menviel et al., modelling study is mostly forced with changes in the wind strength. Whilst I agree the modelling agrees well with the data and supports the hypothesis that the solubility pump was important during the deglacial CO₂ rise, it doesn't necessary follow that it rules out biological pump. For that, one would need to run an experiment with both changes in winds and dust delivery and show disagreement with the data.

We agree with the reviewer that the modelling of Menviel et al. (2018) cannot be used to rule out any biological pump effect. However, as the reviewer pointed out, our intention to use the data-model comparison is to support the important solubility pump contribution. In the revision, we provide more descriptions of the model simulation (such as the main forcing mentioned by the reviewers, Lines 196-200) and rewrite Discussion to include the caveat raised by the reviewers to clarify the main purpose of the data-model comparison (i.e., to evaluate the solubility pump effect) (Lines 203-206).

This caveat is now included in the manuscript and clearly discussed.

Can the authors be sure what's going on in HS1 without any data in the LGM to give context?

A few times in the paper, the authors make qualitative and quantitative references to changes in the records relative to LGM. However, their main records only extend to 18ka, which is essentially the start of the deglaciation with respect to changes in carbon cycle. This is further compounded by high uncertainty in age model. Can the author demonstrate that their records clearly fall outside the HS1 and thus provide sufficient coverage in the LGM?

The author state age uncertainty "(1-sigma ranging from 0.3 to 1.7 ka) is derived from the Undatable script, and for published records, a uniform age uncertainty (1-sigma) of 0.5 ka is assigned. With an oldest date of 18ka, it seems about equally likely that the oldest date point falls within HS1 rather than the LGM.

We understand the reviewers' concern about our age model and data representation for the LGM. To address the reviewers' questions, we have updated our age model with additional planktic radiocarbon dates during 12.8-18.0 ka BP. With the updated age model, chronology uncertainty of our record is now reduced to 0.2 to 0.8 ka. In particular, the new age model put firmer constraint on the two age points within the LGM by reducing their age uncertainties to 0.3 ka (see supplementary table S1), providing stronger support for using ~18 ka samples to reconstruct LGM conditions. Furthermore, benthic oxygen isotopes show consistent structures to the global benthic oxygen isotope stack (the LR04 curve, Lisiecki and Raymo (2005)), demonstrating that the earliest part of our record lies within the LGM (Fig. X2).

Fig. X2, Benthic oxygen isotope record at site MD97-2106 (red) compared to the global benthic oxygen isotope stack (LR04 (Lisiecki & Raymo, 2005), grey). Grey bars represent the Younger Dryas and Heinrich Stadial 1.

The addition of the new radiocarbon dataset has reduced the error in the age model. The use of 18 ka samples does still seem pretty close to be capturing some of the deglacial change in the samples,

but, at least with 0.3 ka error it seems slightly more assured as the LGM, as opposed to 0.5 ka error as in the previous manuscript.

The response to the comment below, where the authors run a sensitivity test on the reference age, does come a way to meeting these concerns. This would be beneficial to other readers if included in the supplementary information – which I can see the authors have already done.

Even if so, there remains an issue that they have some few data points it will be very difficult to constraint the background LGM conditions. This appears to be crucial to the conclusions of the paper as their deconvolution method requires “We first convert surface-water pCO₂ to relative changes (ΔpCO_2) to a reference age (here chosen as the oldest age of the record).”

The reviewers’ comment can be divided into two parts.

1. Does the deconvolution method depend on the choices of reference age?
2. How well do our data reflect the deglacial change?

In response to part 1, we demonstrate that various deconvoluted seawater pCO₂ changes (e.g., δpCO_2^{BIO} and δpCO_2^{SOL}) marginally depends on the choice of the reference age. This is demonstrated by the application of our method to site PS2498-1 (Main text, Fig. 3c), which has a reference age of 15.8 ka (i.e., the late HS1). Despite that the choice of the reference age (in essence, surface-water pCO₂ of the reference age) would affect absolute values of calculated pCO₂^{SOL} and pCO₂^{BIO}, relative changes in pCO₂^{SOL} and pCO₂^{BIO} (i.e., δpCO_2^{SOL} and δpCO_2^{BIO}) are very similar regardless of the choice of the reference age (Fig X3 also Fig S6).

In response to Part 2, our data at site MD97-2106 is representative of the full deglacial change compared to the LGM condition. The starting point of the record affects the context where the relative change we reconstructed can be interpreted. As the starting point of our record at site MD97-2106 does lie within the LGM as we demonstrated in the response to the last comment, δpCO_2^{SOL} and δpCO_2^{BIO} at site MD97-2106 is representative of the full deglacial change. We do agree with the reviewers that PS2498-1 records are too short to represent the full deglacial changes, which is now acknowledged in the main text now (Line 188).

Fig. X3, Sensitivity tests of reference age effect using MD97-2106 data. **a**, $\delta p\text{CO}_2^{\text{SOL}}$. **b**, $\delta p\text{CO}_2^{\text{BIO}}$. The solid curves show calculated values using a reference age of 18 ka (shown in the main text), while dashed curves show calculations using a reference age of 4 ka. To facilitate comparison between $\delta p\text{CO}_2$ decompositions between methods, calculated $\delta p\text{CO}_2^{\text{SOL}}$ and $\delta p\text{CO}_2^{\text{BIO}}$ using a reference age of 4 ka are further converted to the relative changes to 18 ka.

Please see above comment in response to this.

Polar front shifting and local – regional variability effects on records

There appears to be limited accounting for variability in the records which would be caused by factors external to a broad biological versus solubility pump question. For example, throughout the time-period of the new record (8-20 ka BP), might it have been expected that the polar front would shift latitudinally? How might changes in the waters overhead of the location of the MD97-2106 core affect what we see in the record? For example, the biologically most productive waters could have migrated away from the region recorded by the MD97 core, such that the record indicates low biological pump contribution, whereas in fact the processes on the scale of the whole sub-Antarctic zone (SAZ) were the same, but at different latitudes? In this case, overall sub-Antarctic contribution to global atmospheric CO₂ may have been unchanged, though the MD97-2106 record may have instead been showing changes in pCO₂ processes. Introducing the other core records doesn't necessarily cover this question, since they are found at similar latitudes. Could you give more discussion on how this influences records throughout the wider SAZ? Is it possible to quantify any effect of latitudinal shifts in the records?

This is another very insightful comment. We thank the reviewer for raising the issue of the potential influence of deglacial front migration on our interpretation. Firstly, we stress that when we partition surface-water $\delta p\text{CO}_2$ into $\delta p\text{CO}_2^{\text{SOL}}$ and $\delta p\text{CO}_2^{\text{BIO}}$, influences of frontal shifts have already been

considered, rather than being external of our methodology. This is because 1) $\delta p\text{CO}_2^{\text{SOL}}$ and $\delta p\text{CO}_2^{\text{BIO}}$ reflect SST/SSS changes and carbonate chemistry redistribution, respectively, and 2) the frontal shift would affect surface-water $p\text{CO}_2$ via SST/SSS and carbonate chemistry. We add this point to the main text to avoid any confusion (Lines 134-137).

However, the reviewers' point is insightful in that frontal shifts can affect how $\delta p\text{CO}_2$ is divided into $\delta p\text{CO}_2^{\text{SOL}}$ and $\delta p\text{CO}_2^{\text{BIO}}$. The Southern Ocean fronts generally shifted poleward during the last deglaciation, interrupted by a reversed change during the ACR (Buizert et al., 2018). There are two scenarios where the frontal shift affects $\delta p\text{CO}_2$ decomposition at a site:

1. when a front (Subtropical Front (STF) or the Subantarctic Front (SAF)) sweeps over a site;
2. when the relative position of the site within the SAZ changes.

The influence of frontal shifts on SST and carbonate chemistry at a site would be large when a front sweeps over the site (in Scenario 1). In comparison, if a site remains in the SAZ, the influence of frontal shift, caused by changes in the relative position of a site within the SAZ (Scenario 2), would be much smaller, due to the relative narrow latitude range of the SAZ (Fig. 1 in main text). We state that front did not move over site MD97-2106 during the last deglaciation, based on SST variability of $\sim 3^\circ\text{C}$. Because if the STF, that is associated with a 4°C temperature gradient in the modern ocean, moved over our site, a larger SST change would be expected. This like happened at site TAN1106-28 with a deglacial SST change of $\sim 7^\circ\text{C}$. Our site is suitable for the deglacial SAZ reconstruction, in part because the impact of frontal shift on SST and carbonate chemistry is relatively small.

Nevertheless, as long as there are front shifts, the relative position of any site slightly changes as the reviewers pointed out (Scenario 2). Such a shift would have an impact on biological pump observed at any given site. Poleward frontal shifts reduce nutrient supplied to a site, and thus tend to lower $\delta p\text{CO}_2^{\text{BIO}}$ with everything else being equal. To test if the changes we see at site MD97-2106 is widespread in the SAZ or regional, we introduce other SAZ sites for a more comprehensive view of the whole SAZ. The compiled SAZ sites likely have a good latitude coverage of the SAZ, despite that these compiled sites come from a similar latitude range. This is because 1) the Atlantic SAZ is at slightly lower latitudes than the Pacific SAZ (Fig. 1 in main text), and 2) site PS2498-1 seems to bear some Antarctic Zone character during the last deglaciation (Martinez-Boti et al., 2015) suggesting that it is at the southern end of the SAZ, while MD97-2106 is currently at the northern end of the SAZ. The results of the site compilation differ between $\delta p\text{CO}_2^{\text{SOL}}$ and $\delta p\text{CO}_2^{\text{BIO}}$. We see consistent $\delta p\text{CO}_2^{\text{SOL}}$ changes in all these sites, suggesting that the solubility pump changes are indispensable for the deglacial surface-water $p\text{CO}_2$ rise in the SAZ. By contrast, the deglacial $\delta p\text{CO}_2^{\text{BIO}}$ variabilities at

these sites differ, likely in response to varying relative contributions of front-related nutrient supplies and iron-related nutrient utilization.

The reviewers also point out that position of the SAZ shifted together with the fronts, so that observed changes at the compiled SAZ sites may not represent changes of the whole SAZ. We fully agree with the reviewers about implications of frontal shifts on our data interpretation. We now acknowledge the strong influence of frontal shifts on $\delta p\text{CO}_2^{\text{BIO}}$ in our interpretation (Lines 220-228). However, $\delta p\text{CO}_2^{\text{BIO}}$ at examined sites is also determined by iron-induced nutrient utilization changes with an opposing effect on $\delta p\text{CO}_2^{\text{BIO}}$ to frontal shift during deglaciations. As a result, a more comprehensive consideration of our data following the reviewers' comments real that our data implies an iron-induced nutrient utilization changes in the SAZ.

We agree with the reviewers on the deglacial shift of the SAZ itself, and acknowledge this point in the main text (Lines 222-223). When evaluating changes of $\delta p\text{CO}_2^{\text{SOL}}$ and $\delta p\text{CO}_2^{\text{BIO}}$ in the entire SAZ, the best we can do in a work based on paleo-reconstructions is to compile all the similar records in the region. We note that in paleo modelling studies, the SAZ can only be coarsely resolved in most of the current earth system models, and the meridional shift of the SAZ would be challenging to pin down with great accuracy. We concur with the reviewers that a more comprehensive view on the SAZ's role in contributing to the deglacial carbon cycle change should consider shifts of the SAZ, which warrants more paleo-reconstructions and modelling works.

The authors have taken the time to fully consider and communicate the frontal shift effects in response to the review comments, and have incorporated this into the manuscript. While the thinking behind this is very clear in this reply, I would like to suggest clarifying this in the discussion section of the main manuscript, as it is less clear here.

The below is the main confusions for me;

In L226-227 the discussion reads that 'our reconstructions do not exclude a potential role of iron fertilisation', and here the paragraph ends. The following paragraph expands on the observed influence of the solubility pump. The next paragraph then jumps to the conclusion 'In sum' that (L246) 'these reconstructions imply iron-related nutrient utilisation effects on deglacial SAZ'.

The flow of this needs to be improved such that this growth in idea is explained in the discussion in a consistent way. This should be easily done by changing the paragraph structure of the discussion and including some of the detail included in the above comment reply.

One other minor comment on this reply; 'This like happened at site TAN1106-28 with a deglacial SST change of $\sim 7^\circ\text{C}$,

Can you include this in the main text where it would be most appropriate to include it around the discussion of Fig 3? This would help interpretation of why this site looks quite different to the others, alongside the lower resolution data.

Similarly – how much might the spatial variability of sub-Antarctic waters and productivity affect interpretation of the records? In Figure 3, the authors show records from three other sites, and use them to confirm the contribution of the solubility effect throughout the sub-Antarctic (L139-155). However, there are also some large differences between the records throughout different time periods. 15N records from the two cores MD97-2106 and ODP-1090 are, for example, quite different. In MD97-2016, changes in 15N seem to correspond to changes in $p\text{CO}_2^{\text{bio}}$, but in ODP1090 the steep Holocene change in 15N does not seem to be reflected in the $p\text{CO}_2$ record. How can these differences be reconciled with the overall conclusions of the paper? Can you offer some more discussion of different scales of variability in considering surface water $p\text{CO}_2$ levels in the sub-Antarctic?

As the reviewer mentioned, we introduce the other SAZ records mainly to confirm the contribution of the solubility pump throughout the SAZ. The reviewers also spot the complexity associated with productivity changes that affect both $\delta^{15}\text{N}$ and $\delta p\text{CO}_2^{\text{BIO}}$, which we now discuss in more details (Lines 147-151, 220-228). We have provided more detailed explanation of the $\delta p\text{CO}_2^{\text{BIO}}$ in the main text as included in last response. For foraminifer-bound $\delta^{15}\text{N}$, although it is often used to reflect nutrient utilization efficiency, it is also affected by $\delta^{15}\text{N}$ of the nitrate supplied to the SAZ (Martínez-García et al., 2014; Ren et al., 2012; Ren et al., 2015; Wang et al., 2017). The deglacial part of site ODP 1090 has been thoroughly discussed previously and was interpreted to be affected by both the nitrate utilization and source $\delta^{15}\text{N}$ (Martínez-García et al., 2014). Martínez-García et al. (2014) argue that “*a poleward migration of the Subantarctic Front upon deglacial warming would decrease the concentration and increase the $\delta^{15}\text{N}$ of the nitrate being supplied to Site 1090, both of which would work to raise FB- $\delta^{15}\text{N}$ at a given rate of productivity and nitrate use*”. We decide not to go into these details to explaining $\delta^{15}\text{N}$ at our site, because similar reasoning have been provided by previous studies, and our focus is $\delta p\text{CO}_2^{\text{BIO}}$ rather than $\delta^{15}\text{N}_{\text{FB}}$. For $\delta^{15}\text{N}_{\text{FB}}$ at site MD97-2106, we stress that its similar structure to independently derived $\delta p\text{CO}_2^{\text{BIO}}$ indicates that the $\delta^{15}\text{N}_{\text{FB}}$ is likely a good indicator for nutrient utilization at this site, despite the abovementioned complexities. The inter-basin differences in $\delta^{15}\text{N}_{\text{FB}}$ and $\delta^{15}\text{N}-\delta p\text{CO}_2^{\text{BIO}}$ relation, which are potentially linked to zonal differences in

relative contributions of local nutrient utilization and nutrient supplies related to frontal shifts. Nevertheless, we deem that this is outside the scope of this manuscript and warrant further studies focusing on the inter-basin difference in the SAZ biological pump. This is because our focus, as the reviewers agreed with, is the widespread solubility pump's contribution to surface-water pCO₂ in the SAZ during the last deglaciation.

The edits made within the manuscript satisfactorily meet the questions in this comment.

In considering the age scale of the record – the authors say that the age model of MD97-2106 is based on tuning of sea surface temperature (SST) of the new record to Antarctic temperature from the EPICA Dome C ice core record. Would the timing of SST temperature change in the sub-Antarctic be the same as temperature changes of the central Antarctic continent, allowing this ‘wiggle match’ dating technique? Unless the two regions experience change at the same time, then the timescale of the MD97 record may be incorrectly offset. Does the calculated error for the timescale include this potential offset?

For the age model, we simply adopt the method that is widely used in the paleo community. This is based on Matching SST at our site with EDC δD for the deglacial age model takes the advantage of the synchronized mid-high latitude SST in the Southern Hemisphere and Antarctic temperature (e.g.,(Pedro et al., 2015)), and a similar method have been applied to other Southern Ocean records (e.g.,(Waelbroeck et al., 2019)). Moreover, our updated age model (see replies to a previous comment) shows that the SST- δD tie points agree with radiocarbon dates during the last deglaciation. To demonstrate the reliability of our age model, we tested the chosen age model against two alternative age models based on 1) only radiocarbon dates and 2) only SST- δD tie-points during the last deglaciation (Fig. X4, also Fig S4). As shown in Fig. X4, these age models, with discrepancies smaller than 0.8 ka show consistent structures for Mg/Ca-based SST, and oxygen isotopes.

Fig. X4, Comparisons of age models based on only radiocarbon dates (**a**) and only SST- δD tie-points (**b**). At the bottom of the two panels, crosses and triangles represent tie-points based on radiocarbon and SST- δD tie-points, respectively. To illustrate the consistency between different age models, Mg/Ca-based SST, planktic oxygen isotopes, and benthic oxygen isotopes using varying age models are compared to appropriate reference records.

As in previous comments, the addition of the radiocarbon dating adds to the robustness of the age model. The inclusion of these tests varying the age model shows robustness in the choice of model, and the inclusion of these tests as Supplementary Information is beneficial to the readers.

Why does the deconvolution require a fixed reference point with constant SST and salinity? L289-301

As I read this, the authors fix both temperature and salinity to a single reference point throughout the record in calculating pCO_{2bio} . But I would have thought the biological pump could also be affected by temperature changes?

In their method of calculating pCO_{2sol} by subtracting the found pCO_{2bio} from $pCO_{2in-situ}$, could this artificially enhance (or reduce) pCO_{2sol} by assuming a lower (or greater) pCO_{2bio} ?

More to the point, their record of SST clearly shows changes in temperature, thus this assumption of constant temperature seems to be invalidated. As a reader, the description was a bit too vague to understand the ramifications. Can you please expand on this assumption of reference to a constant temperature and salinity?

We understand the reviewers' concern about the method used to decompose surface-water $\delta p\text{CO}_2$, a point also raised by reviewer 3. In the revision, we use a more straightforward method for our calculations with step-by-step descriptions (Lines 123-137), and leave the method from the previous version only in the Methods and supplementary materials. Here, we will first re-quote some part of the comment to explain the rationale behind our decomposing methods. Then, we will re-quote and respond to the rest of the comment.

The methods now included in the manuscript are much clearer than the previous version.

Why does the deconvolution require a fixed reference point with constant SST and salinity?

More to the point, their record of SST clearly shows changes in temperature, thus this assumption of constant temperature seems to be invalidated.

These comments are related to the rationale of our method.

Based on its definition (Zeebe & Wolf-Gladrow, 2001), seawater $p\text{CO}_2$ is determined by carbonate chemistry (DIC and alkalinity) and physical properties (temperature, salinity, and pressure; in the case of the surface seawater, pressure is constant). In all the natural settings, including both modern and paleo settings, carbonate chemistry will never be constant while SST/SSS changes, nor will SST/SSS remain constant while carbonate chemistry changes. Our methodology seeks to attribute relative surface-water $p\text{CO}_2$ changes at each site to changes in the biological pump and the solubility pump. The biological pump component ($\delta p\text{CO}_2^{\text{BIO}}$) corresponds to surface-water $p\text{CO}_2$ change caused by redistribution of ALK and DIC (mainly DIC), while the solubility pump component ($\delta p\text{CO}_2^{\text{SOL}}$) corresponds to surface-water $p\text{CO}_2$ change caused by SST and SSS changes (mainly SST). To achieve this goal, i.e., to derive the surface-water $p\text{CO}_2$ changes caused by one pump, parameters related to the other pump must be *hypothetically* fixed. Because we attribute the $\delta p\text{CO}_2$ to either biological or solubility components, we can derive the component of one of the pumps directly and attribute the remainder to the other pump.

In this new version, we derive the solubility pump component directly, and attribute the remainder of $\delta p\text{CO}_2^{\text{TOTAL}}$ to the biological pump. This method, we believe, is more straightforward to readers, and it is similar to that is routinely used in analysing modern ocean carbonate chemistry data (e.g., (Nicholson et al., 2022)). We also provide results derive the biological pump component directly (i.e.,

the method in the previous version) in the supplementary material. These two methods generate similar results (Fig X5, also Fig S8), providing strong support to our conclusion.

We also provide explanation of the rationale of the method and a step-to-step description of the new method in the main text (Lines 123-137), and we also include a step-to-step description of the other method in the Methods section (Lines 396-408).

Fig. X5, $\delta p\text{CO}_2^{\text{TOTAL}}$ partition at site MD97-2106 using two methods. **a,** $\delta p\text{CO}_2^{\text{SOL}}$. **b,** $\delta p\text{CO}_2^{\text{BIO}}$. The solid curves show calculations presented in the main text using a $\delta p\text{CO}_2$ -partition method deriving $\delta p\text{CO}_2^{\text{SOL}}$ directly, while the dashed curves show calculations using an alternative $\delta p\text{CO}_2$ -partition method deriving $\delta p\text{CO}_2^{\text{BIO}}$ directly.

Ok, this explanation clarifies the use of fixed temperature and it is also now included in the new methods section in the manuscript. The inclusions of this further investigation comparing method outputs in the supplementary information is again useful.

In their method of calculating $p\text{CO}_2^{\text{sol}}$ by subtracting the found $p\text{CO}_2^{\text{bio}}$ from $p\text{CO}_2^{\text{in-situ}}$, could this artificially enhance (or reduce) $p\text{CO}_2^{\text{sol}}$ by assuming a lower (or greater) $p\text{CO}_2^{\text{bio}}$?

This comment is technical about our method. Although we now derive $\delta p\text{CO}_2^{\text{SOL}}$ first, we face a similar issue. In the case of the new method, the recalculated $\delta p\text{CO}_2^{\text{SOL}}$, to a small degree, depends on the choice of reference age, in essence surface-water $p\text{CO}_2$ of the reference age, because the response of seawater $p\text{CO}_2$ to temperature is exponential, which we now state clearly in the Methods (Lines 406-408). This issue has been discussed at length in a response to a previous comment. In short, by only investigating the relative changes ($\delta p\text{CO}_2^{\text{SOL}}$ and $\delta p\text{CO}_2^{\text{BIO}}$) compared to 18 ka, the absolute levels of these two components do not matter much (Fig. X3, also Fig S6).

This comment has been answered by both authors and reviewer response to authors in the above sections.

L289-301 As I read this, the authors fix both temperature and salinity to a single reference point throughout the record in calculating pCO_2^{bio} . But I would have thought the biological pump could also be affected by temperature changes?

We define the biological pump component (δpCO_2^{BIO}) as the surface-water pCO_2 change caused by redistribution of ALK and DIC (mainly DIC), while the solubility pump component (δpCO_2^{SOL}) corresponds to surface-water pCO_2 change caused by SST and SSS changes (mainly SST). Consequently, following our method, the effect of biological pump change is not related to temperature, but only changes in DIC and ALK.

Thank you for clarifying this.

Minor comments:

Capitalisation of the subantarctic/subantarctic appears inconsistent.

We now use the capitalized Subantarctic consistently in the reversion.

L55: urging a rethinking of mechanisms

L83: tuning

The above errors are corrected.

L83: recommend adding comma after (SST), and site, to improve readability of the sentence

The sentence is rearranged for clarity (Lines 101-102).

L93: age model of our core

Corrected.

L96-98: can you be clearer on the latter part of this sentence? But....(what?) might be complicated by?

This sentence is removed from the reversion.

L115: caused by, instead of due to

L119: Should it be 'we noted'? Not sure what 'It' is

The above errors are corrected.

L119-123: Long sentence, suggest breaking up

The sentence is rewritten for clarity (Lines 147-149).

L138: ...pCO₂ changes 'in the area represented by our record' from the...or something similar would be clearer here

We change it to the Pacific SAZ (Line 172).

L141-143: Not clear on what is referred to here. In Fig.3 it looks like the solubility range is approx.0-40ppm; where is the 40-90ppm from?

It should be 40-60 ppm. We refer to the variabilities (the maximum minus the minimum) of δpCO_2^{SOL} at these sites. This sentence is now removed in this version.

L143: capitalise South Atlantic

L146: despite pCO₂bio dictating

L150: when most of the deglacial

The above errors are corrected.

L168: reducing iron deposition? Or limited?

Changed to reducing (Line 211).

L172-173: It seems an over statement to say 'most' deglacial pCO₂ rise was caused by solubility – results in Fig 3 for example don't show that solubility was consistently explaining most of the pCO₂ record in comparison to biology, but it was a contributor

We have removed this sentence.

Fig 2: Could the shaded bars be labelled on the figure?.

Fig 3: Shaded bars are not defined here (same in extended data figures 3 and 5)

All the shaded bars are defined and labelled.

L235/237: can you define ICP-MS and MC-ICP-MS on first use?

We have defined these terms (Lines 334, 336).

L239: why is 'instead of NaAc-HAc mixture' significant, can you explain?

We have explained the reason in the next sentence (Line 340).

You switch back and forth between using boron and B, more consistent to define boron as B and use it, or just use boron.

We now use "boron" consistently.

L243: in to micro-columns? Or on to microcolumns?

We have removed this phrase.

L246: the resin eight times

L247: five aliquots

L250: at ANU

L306: to the solubility pump

L318: in an 80C bath

L321: into a combusted

The above errors are corrected.

L322-325: would be clearer to say the reagent was added to samples

We have changed the sentence following the reviewers' suggestion (Line 438).

L335: could you be more specific than 'generally'? This word seems open to wide interpretation

We have removed the word "generally".

Further comments:

Fig 2, a/b, d/e: It is not stated which line on the plot is represented by which axis

L64: the word carbon is repeated

L96/97: to the south of our site

L178: Can you offer any suggestion as to why there is the excursion in pCO₂Bbio at 17 ka in ODP 1090? Or comment on if this is within the realms of expected local-scale variability?

L220: Having said that two of your extra sites show no correlation with opal/lithogenic flux, could you mention that the third site does not have an opal/flux record just to be clear on why this relationship is not said to be observed for all the sites (as opposed to allowing the reader to make the assumption instead that this record may have shown an opposite trend, for example).

Fig 4: Can you add the different core locations for reference, and would it be easily possible to generate these figures on geographical scales such that they are comparable to Fig 2?

L370: Define N-TIMS

We would thank the reviewers again for the very constructive reviews that greatly help us improve the manuscript. Below, we copy the reviewers' new comments (*grey and italic*) and provide responses to comments where appropriate.

The authors have paid close attention to the reviewer comments and have answered them in detail with considered answers that will have taken both time and thought. They have taken our comments on board in edits to the manuscript. Overall, the changes made to the manuscript have clarified and improved the text greatly. Both the methods and impact of the findings are more directly communicated, without the latter being (previously probably unintentionally) overamplified. This is a robust study, and the conclusions are of significance.

Below are responses to each of the authors answers and edits, with a few minor suggestions for further improvement.

This consideration of the main questions of the study is valuable, and as the revised manuscript reads these are now much more apparent and consistent throughout.

'We hope the reviewers agree with us that addressing Question #2 warrants separate studies.'

This is reasonable, given the improved communication of the new text providing a clearer picture of the questions you aim to answer in the study. The omission of the need to investigate the global contribution further does not take away from the study giving conclusions of significance. Rather the findings that are presented now appear improved in their significant via clearer emphasis.

We thank the reviewer for the positive review.

Ok, this is clarified.

Again, the refocussing of the text on the SAZ specifically has presented a clearer story and negated the need to calculate global impacts, while not removing the significance of the findings. It is now reasonable to state that more detail on the global impacts to the carbon budget are beyond the scope of the study. However, with the original comment in mind, it still doesn't seem quite right to say the solubility pump has been over-looked (L13, abstract). It has been included in these previous modelling studies, highlighted by all reviewers. Rather, it seems that it is the significance or potential contribution that has perhaps been underestimated. Or, do you say this for studies focussing on the SAZ very specifically? Could the authors re-phrase slightly to reflect whichever case?

We intended to specifically mean the solubility pump effects in the SAZ. We added "regional" (Line 12) in this sentence for clarification.

The nomenclature is clarified and consistent throughout the manuscript. Just on a technicality—should the p of $p\text{CO}_2$ be italicised.

Both $p\text{CO}_2$ (e.g., in Zeebe and Wolf-Gladrow (2001)) and $p\text{CO}_2$ (e.g., in Sarmiento and Gruber (2006)) are used in the literature. We double-checked our manuscript to ensure that $p\text{CO}_2$ is consistently used.

This caveat is now included in the manuscript and clearly discussed.

The addition of the new radiocarbon dataset has reduced the error in the age model. The use of 18 ka samples does still seem pretty close to be capturing some of the deglacial change in the samples, but, at least with 0.3 ka error it seems slightly more assured as the LGM, as opposed to 0.5 ka error as in the previous manuscript.

The response to the comment below, where the authors run a sensitivity test on the reference age, does come a way to meeting these concerns. This would be beneficial to other readers if included in the supplementary information – which I can see the authors have already done.

Please see above comment in response to this.

The authors have taken the time to fully consider and communicate the frontal shift effects in response to the review comments, and have incorporated this into the manuscript. While the thinking behind this is very clear in this reply, I would like to suggest clarifying this in the discussion section of the main manuscript, as it is less clear here.

The below is the main confusions for me;

In L226-227 the discussion reads that ‘our reconstructions do not exclude a potential role of iron fertilisation’, and here the paragraph ends. The following paragraph expands on the observed influence of the solubility pump. The next paragraph then jumps to the conclusion ‘In sum’ that (L246) ‘these reconstructions imply iron-related nutrient utilisation effects on deglacial SAZ’.

The flow of this needs to be improved such that this growth in idea is explained in the discussion in a consistent way. This should be easily done by changing the paragraph structure of the discussion and including some of the detail included in the above comment reply.

In order to be consistent with discussion here, we revise the first part mentioned by the reviewer (Line 226-227 in the previous version) to state that our reconstructions imply an iron-related influence on surface-water $p\text{CO}_2$ (Line 233-235). We also expanded on how the iron-induced nutrient utilization and frontal shifts could have opposing effects on $\delta p\text{CO}_2^{\text{BIO}}$ (Line 229-235).

One other minor comment on this reply; ‘This like happened at site TAN1106-28 with a deglacial SST

change of ~7 C',

Can you include this in the main text where it would be most appropriate to include it around the discussion of Fig 3? This would help interpretation of why this site looks quite different to the others, alongside the lower resolution data.

We incorporate this point when discussing TAN1106-28 data in the main text (Line 190).

The edits made within the manuscript satisfactorily meet the questions in this comment.

As in previous comments, the addition of the radiocarbon dating adds to the robustness of the age model. The inclusion of these tests varying the age model shows robustness in the choice of model, and the inclusion of these tests as Supplementary Information is beneficial to the readers.

The methods now included in the manuscript are much clearer than the previous version.

Ok, this explanation clarifies the use of fixed temperature and it is also now included in the new methods section in the manuscript. The inclusions of this further investigation comparing method outputs in the supplementary information is again useful.

This comment has been answered by both authors and reviewer response to authors in the above sections.

Thank you for clarifying this.

Further comments:

Fig 2, a/b, d/e: It is not stated which line on the plot is represented by which axis

This is now clarified in the figure caption.

L64: the word carbon is repeated

Fixed.

L96/97: to the south of our site

Fixed.

L178: Can you offer any suggestion as to why there is the excursion in pCO₂Bbio at 17 ka in ODP1090? Or comment on if this is within the realms of expected local-scale variability?

We think this $\delta p\text{CO}_2^{\text{BIO}}$ excursion is likely due to local-scale nutrient supplies, as discussed in Shuttleworth et al. (2021) for surface-water $p\text{CO}_2$. We therefore decided not to discuss this excursion in detail in this manuscript.

L220: Having said that two of your extra sites show no correlation with opal/lithogenic flux, could you mention that the third site does not have an opal/flux record just to be clear on why this relationship is not said to be observed for all the sites (as opposed to allowing the reader to make the assumption instead that this record may have shown an opposite trend, for example).

We now state that the opal and lithogenic flux data is available at the two mentioned sites (Line 223-225).

Fig 4: Can you add the different core locations for reference, and would it be easily possible to generate these figures on geographical scales such that they are comparable to Fig 2?

Good point. We changed the figure (adding core locations and changed the geographical scales) following the reviewer's advice.

L370: Define N-TIMS

Defined.

References

- Sarmiento, J. L., & Gruber, N. (2006). *Ocean biogeochemical dynamics*: Princeton University Press.
- Shuttleworth, R., Bostock, H. C., Chalk, T. B., Calvo, E., Jaccard, S. L., Pelejero, C., et al. (2021). Early deglacial CO_2 release from the Sub-Antarctic Atlantic and Pacific oceans. *Earth and Planetary Science Letters*, 554. 10.1016/j.epsl.2020.116649
- Zeebe, R. E., & Wolf-Gladrow, D. (2001). *CO_2 in Seawater: Equilibrium, Kinetics, Isotopes: Equilibrium, Kinetics, Isotopes*: Elsevier.